# Prevention of Food Allergy: The Significance of Early Introduction

**DOI:** 10.3390/medicina55070323

**Published:** 2019-06-30

**Authors:** Pasquale Comberiati, Giorgio Costagliola, Sofia D’Elios, Diego Peroni

**Affiliations:** Department of Clinical and Experimental Medicine, Section of Paediatrics, University of Pisa, 56126 Pisa, Italy

**Keywords:** allergy, breastfeeding, children, complementary foods, food allergy, microbiome, tolerance

## Abstract

Over the last two decades, the prevalence of food allergies has registered a significant increase in Westernized societies, potentially due to changes in environmental exposure and lifestyle. The pathogenesis of food allergies is complex and includes genetic, epigenetic and environmental factors. New evidence has highlighted the role of the intestinal microbiome in the maintenance of the immune tolerance to foods and the potential pathogenic role of early percutaneous exposure to allergens. The recent increase in food allergy rates has led to a reconsideration of prevention strategies for atopic diseases, mainly targeting the timing of the introduction of solid foods into infants’ diet. Early recommendation for high atopy risk infants to delay the introduction of potential food allergens, such as cow’s milk, egg, and peanut, until after the first year of life, has been rescinded, as emerging evidence has shown that these approaches are not effective in preventing food allergies. More recently, high-quality clinical trials have suggested an opposite approach, which promotes early introduction of potential food allergens into infants’ diet as a means to prevent food allergies. This evidence has led to the production of new guidelines recommending early introduction of peanut as a preventive strategy for peanut allergy. However, clinical trials investigating whether this preventive dietary approach could also apply to other types of food allergens have reported ambiguous results. This review focuses on the latest high-quality evidence from randomized controlled clinical trials examining the timing of solid food introduction as a strategy to prevent food allergies and also discusses the possible implications of early complementary feeding on both the benefits and the total duration of breastfeeding.

## 1. Introduction

Food allergy (FA) has now become a significant pediatric health issue in many developed countries worldwide [1,2]. Although accurate epidemiological data are lacking, there is a strong impression that FA prevalence has significantly increased over the last two decades in Westernized societies, where rates as high as 10% have been properly documented among preschool children [3]. In these countries, pediatric allergists are also experiencing significant changes in the pattern of allergic sensitization and FA manifestations, with a wider range of allergenic foods and severity of reactions [4]. Notably, a significant increase in the prevalence of FA has recently been documented also in regions with rapid economic growth and urbanization on a massive scale, such as China [5].

The rising prevalence of FA has led to a reconsideration of primary prevention strategies related to dietary and nutritional interventions. International recommendations of the early 2000s to delay the introduction of potential food allergens after the first year of life in atopy prone infants have been rescinded, as emerging evidence has shown that this approach is not effective for FA prevention [6]. Recent advances in understanding the pathogenesis of FA have promoted ongoing research into the identification of modifiable environmental factors that could be targeted for the primary prevention of FA. In the last decade, this area of research has primarily focused on the timing of the introduction of solid foods to infants’ diet and has provided evidence to suggest that early ingestion of potential food allergens could be an effective preventive strategy for FA [6].

This review addresses high-quality evidence from randomized controlled trials (RCTs) conducted in the last 10 years examining the timing of solid food introduction in early life as a mean to prevent FA and also discusses the possible implications of early complementary feeding on both the benefits and the total duration of breastfeeding.

## 2. Advances in the Pathogenesis of Food Allergy

The molecular and cellular mechanisms involved in the development of food tolerance and how alterations in these mechanisms might promote the development of FA are very complex and extensively reviewed elsewhere [7,8,9]. Although atopic diseases have a significant genetic component [10], the recent rapid increase in FA rates has to be primarily ascribed to environmental factors, which could influence the process of food tolerance either directly or through epigenetic modifications [11,12,13]. Thus far, these environmental drivers have not been completely understood, although the most recent hypothesized mechanisms involve the dual exposure to food allergens, current lifestyle influences on the intestinal microbiome and immune responses (i.e., the hygiene hypothesis), and nutritional imbalances of Western diets with regard to vitamins, antioxidants, fibers and fatty acids contents [11].

According to the “Dual allergen exposure hypothesis”, the initial site of exposure to food allergens during the first year of life could influence the immune system to develop either FA or tolerance. This hypothesis addresses the potential pathogenic role of transcutaneous exposure to food allergens during early infancy while they are not ingested, bypassing intestinal tolerance. This hypothesis is supported by evidence that, in infants with inflamed eczematous skin, the percutaneous exposure to food allergens can promote a Th2 dominant response and consequent allergic sensitization [14], with greater severity of eczema further increasing this risk [15]. Therefore, especially in infants with atopic dermatitis (AD), a strategy of oral avoidance of potential food allergens during early life while having concomitant environmental exposure can increase the risk for developing transcutaneous sensitization and FA.

In recent years, accumulating evidence has shown that the human microbiome, in particular the resident microbiota of the gastrointestinal tract, plays a central role in modulating early host immune system development and therefore potentially modifying the risk for atopic diseases, including FA [16]. The hypothesis linking the rise in FA prevalence to an altered microbiota composition in infancy has been suggested by epidemiological studies, showing that environmental factors increasing microbial exposure in early life (i.e., childhood exposure to farming life, furry animals, unpasteurized milk, and older siblings) were associated with lower risk of atopic disorders [17,18,19,20,21]. On the contrary, early-life exposures to factors which disrupt commensal microbiota or reduce its diversity (i.e., the use of antibiotics, sanitary living and caesarian delivery) correlated with increased risk for allergic diseases [22,23,24]. Insights from recent murine experimental models have further supported the pathogenic role of microbial dysbiosis during early life and have shed light on multiple mechanisms by which intestinal microbiota could affect FA susceptibility [25,26,27]. Of note, recent human studies have provided evidence for an age-dependent correlation between the intestinal microbiota composition and FA occurrence and resolution, suggesting that early infancy is likely the critical window for interventions [28,29,30].

Additional hypotheses address the immunomodulatory role of some nutrients in the diet, in particular of vitamin D and the possible adverse effects of low vitamin D levels on the development of immune tolerance to foods and on the risk for allergic diseases [31,32,33]. However, thus far, studies investigating dietary nutrients, including vitamin D, have yielded conflicting or inconsistent results mainly due to methodological issues [11].

## 3. Delaying the Introduction of Food Allergens: The Former Recommendations

In the early 2000s, the hypothesis that exposure to solid foods in early infancy could increase the risk for allergic sensitization due to the immaturity and permeability of the gut mucosal barrier led to the recommendations that allergenic foods should be avoided during pregnancy and breastfeeding and the introduction of such dietary allergens should be delayed until after the first year of life in high atopy risk infants (i.e., those with ≥1 first-degree family relative with atopic diseases) [34]. In particular, the American Academy of Pediatrics suggested starting complementary feeding after the first six months of life and postponing the introduction of dairy products until after the first year, of egg after the second year, and peanuts, tree nuts, and fish after the third year of life [35]. In 2006, the American College of Allergy, Asthma, and Immunology extended the recommendation for late introduction of potential allergenic solids to all children, independent of their risk of atopy [36].

Despite these strategies of food avoidance, the prevalence of FA continued to rise in Western societies [2]. The lack of solid evidence for these recommendations constituted the rationale for various subsequent observational studies, which showed that delaying the introduction of potential allergenic foods could actually increase the risk of IgE-sensitization and FA [37,38,39,40,41,42,43], in particular for peanut allergy (PA) [39] and egg allergy (EG) [42]. Of note, some of these studies raised the hypothesis that early ingestion of allergenic solids could even be protective against the development of FA [39,41,42,43,44], potentially preventing the dual exposure mechanisms to food allergens [39].

In light of this accumulating evidence, in 2008, the American Academy of Pediatrics updated the earlier recommendations by recognizing that there was no sufficient evidence to support maternal avoidance and delayed the introduction of potential food allergens into infants’ diet as a means for the primary prevention of FA [45].

## 4. Early Introduction of Solid Foods for Food Allergy Prevention: Current Evidence

The lack of efficacy of delayed introduction of dietary allergens on the epidemic burden of FA, combined with new insights into the pathogenesis of FA, has recently shifted the research interest toward prevention by earlier interventions in infants’ complementary feeding practices [46]. In recent years, several clinical RCTs aimed at establishing whether the earlier introduction of potential allergenic solids into infants’ diet, in particular before six months of age, could favor the acquisition of oral tolerance to these foods over the risk of developing allergic sensitization and FA while avoiding them [47,48,49,50,51,52,53,54,55]. These RCTs are summarized in Table 1, and the results are discussed in the following sections by food allergen type.

### 4.1. Multiple Food Allergens

In 2014, the Enquiring About Tolerance (EAT) trial examined the preventive effect of a very early introduction of multiple food allergens (i.e., milk, egg, peanut, fish, wheat and sesame) in an unselected population of exclusively breastfed infants [47]. Participants were randomized to either an early introduction of the allergenic foods between three and six months of life or to exclusively breastfeeding for six months followed by solid food introduction as recommended by international guidelines (Table 1). In the intention-to-treat analysis, the incidence of FA to at least one of the six foods at three years of age was not significantly different between the two groups. However, the per-protocol analysis showed a statistically significant reduction of overall FA (2.4% vs. 7.3%, *p* = 0.01), PA (0% vs. 2.5%, *p* = 0.003; number-needed-to-treat, NNT 40) and EA incidence (1.4% vs. 5.5%, *p* = 0.009; NNT 26) in the early introduction group compared to the control group. Although these findings could suggest a window of opportunity for FA prevention between 3 and 6 months of age, it should be acknowledged that the high rate of non-adherence in the early introduction group (68.1%) constituted an important source of bias in the per-protocol analysis. Of note, the lowest adherence rate was reported for the introduction of egg (43.1%).

An ongoing RCT, Preventing Atopic Dermatitis and Allergies in Children (PreventADALL) [48], is currently evaluating the early and regular use of skin care (i.e., oil-baths and moisturizers cream) and/or the early introduction of peanut, milk, egg, and wheat, at 3–4 months of age complementary to breastfeeding, as strategies for the prevention of AD and FA in the general population. Unfortunately, it is not yet known when the results will be available.

### 4.2. Peanut

Learning Early About Peanut Allergy (LEAP) is the landmark study showing that early introduction of peanut to infants at high risk of FA was effective in reducing the development of PA compared to avoidance [49]. In this study, 640 infants with moderate-to-severe AD and/or EA and peanut skin prick test (SPT) result of ≤ 4 mm, were randomly assigned to either consume at least 6 g of peanut protein per week between 4 and 11 months of age and up to five years of age or avoid peanut for this period. At five years, the intention-to-treat analysis showed a significantly lower prevalence of PA (documented with an oral food challenge) in the intervention group compared with the avoidance group (3.2% vs 17.2%, *p* < 0.001), corresponding to an absolute risk reduction of 14%, a relative risk reduction of 80% and a NNT of 7.1. Of note, this difference was found both in the group of children with initial negative SPT to peanut (1.9% vs. 13.7%, *p* < 0.001) and in those with SPT results of 1–4 mm (10.6% vs. 35.3, *p* = 0.004). Moreover, peanut-specific IgG and IgG4 significantly increased in the active consumption group over time compared with the control group. Finally, the LEAP-On follow-up study showed that children who frequently consumed peanut for the first five years of life reached sustained unresponsiveness to peanut after discontinuing its ingestion for one year compared to the control group [50].

The ongoing Preventing Peanut Allergy in Atopic Dermatitis (PEAAD) trial is currently assessing the preventive effect of peanut ingestion for one year on the occurrence of PA in infants and also children up to 30 months with AD [6].

### 4.3. Egg

Published RCTs addressing the preventive effect of the early introduction of egg have provided mixed and conflicting results, potentially due to differences in the study populations, outcomes, and designs, including the form of egg used (i.e., raw vs. cooked egg) [47,51,52,53,54,55] (Table 1).

The Solid Timing for Allergy Research (STAR) trial was the first double-blind placebo-controlled RCT designed to assess the timing of egg introduction for the prevention of FA [51]. In this study, 86 infants with moderate-to-severe AD, who had never tried egg, were randomly assigned to either receive a daily dose of pasteurized whole raw egg powder or placebo (rice powder) from four to eight months of age. The study failed to demonstrate a significant preventive effect of the intervention at 12 months and was interrupted due to the considerable frequency of allergic reactions to pasteurized egg (31%) in the intervention group [51]. A subsequent RCT, the Hen’s Egg Allergy Prevention (HEAP), enrolled infants from four to six months of age from an unselected population and randomized them to either receive pasteurized raw egg white powder or placebo three times a week for six months. At one year of age, there was no difference in the prevalence of egg specific-IgE or EA between the active and the placebo group, with the intervention group reporting slightly higher rates [52]. Similar findings were recently reported by the Australian Study Starting Time of Egg Protein (STEP), in which 820 infants of atopic mothers were randomized to either consume daily pasteurized whole raw egg powder or placebo from four to six months to 10 months of age, followed by liberalized introduction to cooked egg in both groups [53]. At 12 months, the prevalence of EA and allergic sensitization to egg documented with SPT was not significantly different between the two study groups [53]. In another recent RCT, the Beating Egg Allergy Trial (BEAT), 319 infants with a family history of atopic diseases were randomized to either consume a daily dose of pasteurized whole raw egg powder or placebo from four to eight months of age and to liberalized egg consumption after that age. At 12 months, oral food challenges confirmed that there were no differences in the prevalence of EA between the two study groups, despite significant immunological changes (i.e., a reduction in cutaneous sensitization to egg white and an increase in egg specific-IgG4) were reported by the active group compared with placebo [54].

Finally, the recently published Prevention of Egg Allergy with Tiny Amount Intake (PETIT) trial assessed an early stepwise introduction of a small amount of heated egg combined with optimal treatment of AD as a different approach to reducing the risk of onset of EA. In this study, infants with mild to severe AD, who had never eaten egg, were intensively treated to prevent acute eczematous flare-ups and randomized to either receive a daily dose of 50 mg of heated egg powder (equivalent to 0.2 g of whole egg boiled for 15 min) from six to nine months, increased to 250 mg from 9 to 12 months, or placebo (squash) from 6 to 12 months. In contrast to previously reported results, the PETIT trial found a significant reduction in the prevalence of EA at 12 months in the intervention group compared with placebo (8% vs. 38%, *p* < 0.0001). Of note, the EA was documented by an oral food challenge with “heated” egg powder and not with raw egg as in previously reported studies. A significant increase in egg specific-IgG4 and a significant reduction of IgE to egg were also documented in the active group compared with placebo (*p* < 0.05). However, it should be acknowledged that an intention to treat analysis was not performed in the PETIT study, as 20 (14%) of the 147 infants initially enrolled did not undergo the oral food challenge at 12 months and were excluded from the final analysis [55].

### 4.4. Milk, Cereals, and Fish

Most of the available data on the effect of early introduction of cow’s milk (CM) protein, cereals and fish are provided by observational studies, which showed conflicting results.

A Finnish birth cohort including 6209 infants followed for 18–34 months found that the introduction of CM protein within the first few days of life was associated with an increased risk of developing CM allergy [56]. On the contrary, an Israeli observational birth cohort study found that the exposure to CM protein within the first two weeks of life reduced the risk of subsequent CM allergy, whereas its introduction between four and six months increased this risk [57].

Regarding cereals, a birth cohort study of 1612 infants followed prospectively to almost five years of age found that the introduction of cereals after six months of age increased the risk of parent-reported wheat allergy [58]. Another observational birth cohort of 3781 infants found that the introduction of cereals (i.e., wheat, rye, oats, and barley) at 5–5.5 months of age was associated with a reduced risk of parent reported asthma and allergic rhinitis in their siblings by the age of five years, whereas the introduction of cereals at less than 4.5 months increased the risk of parent reported AD [59]. Similarly, another birth cohort study of 1293 infants showed that first exposure to oat before 5.5 months of age was associated with a reduced risk of parent reported persistent asthma by the age of five years [60]. Unfortunately, the association found in observational studies does not prove a causal relationship. A recent meta-analysis did not provide evidence for a protective effect of early introduction of cereals into infants’ diet for the prevention of FA [61].

There have been many observational studies investigating the relationship between the timing of the introduction of fish and the risk of asthma and atopic diseases, based on evidence of anti-inflammatory properties of omega-3 fatty acids naturally present in high quantities in fish [59,60,62,63,64]. Two recent meta-analyses have pooled these results and concluded that there is low-to-very low evidence to show that early introduction of fish (i.e., before the age of nine months) could reduce the risk of sensitization to any allergens, the development of rhinitis [61], and asthma [65].

Thus far, the only available intervention trial is the EAT study (Table 1), which showed that early introduction (i.e., between three and six months of age) of six allergenic foods, namely CM, wheat, sesame, white fish, peanut and egg, did not reduce the development of FA to these dietary allergens compared with the standard introduction (i.e., after age six months) in the general population [47]. This study had critical issues with protocol adherence, as only 31.9% of thef participants enrolled in the early introduction group adhered to the diet. Adherence was highest for milk (82%), which was recommended as first food in the form of yogurt, and lowest for more textured foods such as fish (60%), sesame (50.7%) and egg (43.1). In addition, yogurt could be directly served, whereas the other foods required preparation such as cooking or pureeing. Finally, the different taste might have played a critical role for some food such as sesame. Taken together, the immaturity of oral motor skills in early infancy along with the logistic demand of introducing several foods with different texture and taste in a few months could in part explain the poor adherence reported in the EAT study. These findings underline the difficulties of such interventions in real life scenarios and suggest that dietary recommendations might achieve greater adherence by focusing on the introduction of fewer food allergens given in liquid form.

## 5. Impact of Early Solid Food Introduction on Breastfeeding

Breast milk is the first feeding source for a newborn, providing nutrients, growth factors, immunomodulatory and anti-inflammatory components, which are crucial for the correct development and health of infants, and possibly influences the development of atopic disorders [66]. Current evidence neither supports nor excludes the potential protective role of breastfeeding on the development of FA [66], however, exclusive breastfeeding is recommended by World Health Organization (WHO), as well as many general pediatric guidelines, until the age of six months [67,68]. These recommendations were not designed for the primary prevention of FA, but to specifically enhance the nutritional benefits and the protective effects of breast milk against gastrointestinal and respiratory infections, which represent the leading causes of death in developing countries [67,68,69].

Recent evidence suggesting that the introduction of allergenic solids before the age of six months might reduce the risk of FA would contradict the WHO recommendations on exclusively breastfeeding. There are limited available data exploring the potential implications of earlier complementary feeding on the total breastfeeding duration, which mainly derive from the EAT [47,70] and LEAP trials [49] and a few observational studies [71,72]. These data suggest that earlier complementary feeding is not associated with a reduced total duration of breastfeeding, supporting the hypothesis that this strategy could coexist with the continuation of breastfeeding. However, at present, there is a lack of data examining the potential impact of early solid food introduction on the numerous beneficial outcomes for infants’ health associated with exclusive breastfeeding, which arguably are more important than FA prevention in developing and poor regions of the world [72].

## 6. Timing of the Introduction of Potential Food Allergens: Current Recommendations

The 2008 guidelines of the American Academy of Pediatrics [45], as well as the 2013 guidelines of the American Academy of Allergy, Asthma and Immunology [73], recommend not delaying complementary feeding after 4–6 months of age. However, none of these guidelines specify the optimal timing of introducing solid foods, and in particular allergenic solids. In 2016, the Australasian Society of Allergy and Clinical Immunology updated their guidelines for FA prevention recommending that complementary feeding should start “around 6 months, but not before 4 months of age“, regardless of a family history of atopy and preferably whilst breastfeeding [74]. Similarly, the 2018 guidelines of the Asian Pacific Association of Pediatric Allergy, Respirology and Immunology recommend the introduction of solid foods, including allergenic solids, from the age of six months in both the general population and infants with a family history of atopic disorders [75].

At present, there is no convincing evidence that early introduction of potentially allergenic foods can prevent FA to the same food, besides the introduction of peanut from 4 to 11 months of age in infants at high risk of developing PA [47]. This evidence led the American National Institute of Allergy and Infectious Diseases to release addendum guidelines for PA in 2017 [76], which recommend peanut introduction as early as 4–6 months in infants with severe AD and/or EA after performing SPT or specific-IgE to peanut. In the case that the SPT result is ≤2 mm (or specific-IgE <0.35 kUA/L), peanut can be introduced at home, whereas, if the result is 3–7 mm (or specific-IgE ≥0.35 kUA/L), an in-office supervised oral peanut challenge is recommended. Infants with SPT result ≥8 mm have a high likelihood of PA and should be managed by a specialist. For infants with mild to moderate AD, these guidelines recommend introducing peanut at around six months, with no need for an in office evaluation. Finally, for all infants with no eczema or FA, peanut can be introduced according to parental preferences and cultural dietary practice. Importantly, for all infants, the exposure to peanut should be attempted by using age appropriate peanut-containing food (such as peanut butter or flour) to avoid the risk of inhalation of peanut kernels. In addition, first introduction of peanut-containing foods should only be attempted after having integrated other nutritionally important foods into the diet, to demonstrate that the infant is developmentally ready for peanut [76].

Currently, there are no specific guidelines for the early introduction of potential allergenic solid foods other than peanut. Although a recent meta-analysis of the five RCTs reported in Table 1 [61] found moderate certainty evidence indicating that egg introduction at the age of 4–6 months reduced the onset of EA, the conclusions were heavily based on the PETIT study [55], in which the stepwise introduction of cooked egg after six months reduced EA to cooked egg. However, in the other four RCTs included in the meta-analysis, the introduction of raw egg before six months did not provide any protective effect on the onset of EA [51,52,53,54]. Thus, the available data may only support the introduction of cooked egg at 6–8 months of age for FA prevention [74,75].

## 7. Conclusions

Maternal avoidance of allergenic solids during pregnancy or breastfeeding and the delayed introduction of such foods in infants’ diet after the first year of life have proven to be an ineffective means of FA prevention and are no longer recommended by international guidelines. Similarly, there is no evidence that very early exposure before four months of age to such dietary allergens can prevent FA both in standard risk and high risk infants. Although recent data suggest considering early introduction of allergenic solids as a potential strategy to tackle the rise in FA prevalence, convincing evidence for such practice is currently available only for peanut in high risk infants between 4 and 11 months of life, but not for most other allergenic foods.

At present, in line with current guidelines [67,74,75], we recommend a progressive introduction to solid foods, including all common allergenic solids, during the first year of life, according to the infant’s ability to chew, keep their head still and sit propped up, and familial or cultural habits, beginning at around six months but not before four months of age, possibly without discontinuing breastfeeding. For infants with severe AD and/or FA, we recommend medical counselling before introducing common food allergens into the diet (i.e., egg, dairy, wheat, fish, and peanut), to exclude an IgE-sensitization to those foods, which would increase the risk of reaction upon ingestion, and to discuss the best timing and modalities of such introductions in light of the available evidence for FA prevention.

There is a need for further well designed RCTs to better understand the potential for FA prevention associated with early introduction of solid foods, the optimal timing of early introduction for each food, and the possible implications of such practices on breastfeeding.

### Key Points:

Maternal avoidance of allergenic solids during pregnancy or breastfeeding and the delayed introduction of such foods in infants’ diet after the first year of life have proven to be an ineffective means of food allergy prevention.Similarly, there is no evidence that intake of allergenic foods, including milk, egg, wheat, fish, and peanut, before four months of age can prevent food allergies in the general population.There is good evidence that, for infants with severe atopic dermatitis and/or egg allergy, regular peanut intake after four months and before 12 months of age can reduce the risk of developing peanut allergy.Available data may only support the introduction of cooked egg at 6–8 months of age to reduce the risk of developing egg allergy.There is a need for more well designed trials to better understand the potential for food allergy prevention associated with early introduction of solid foods other than peanut.

## Figures and Tables

**Table 1 medicina-55-00323-t001:** Summary of RCTs on early introduction of allergenic solids for food allergy prevention.

Trial, Author Year, Country	Study Type	Food Type	Inclusion Criteria	Population (n) Active/Control	Type of Intervention Active vs. Control	Primary Outcome Measurement and Timing
EAT, Perkin, 2014, UK [47]	RCT	Milk (yogurt), peanut, cooked egg, sesame, fish, wheat	General population, 3-month-old term infants, exclusively breastfed	652/651	Sequential introduction of the 6 foods (milk first, the others randomly assigned), 4 g proteins/week, from 3 to 6 months, vs exclusive breastfeeding until 6 months.	Prevalence of allergy to one of the 6 foods by OFC at 1–3 years of age
LEAP, Du Toit 2015, UK [49]	RCT	Peanut (snack/butter)	High risk 4–11-month-old infants, moderate-severe eczema and/or egg allergy. SPT ≤4 mm.	319/321	6 g peanut protein/week, ≥3 times/week, up to 5 years of age vs. peanut avoidance.	Prevalence of peanut allergy by OFC at 5 years of age
STAR, Palmer 2013, Australia [51]	DBPCRCT	Egg (raw whole pasteurized)	High risk 4-month-old infants, moderate-severe eczema (SCORAD score ≥15)	49/37	0.9 g egg protein/daily from 4 to 8 months vs. placebo for 4 months.	Prevalence of EA by OFC with raw egg powder at 1 year of age
HEAP, Bellach 2017, Germany [52]	DBPCRCT	Egg (raw white pasteurized)	General population, 4–6-month-old infants, egg s-IgE <0.35 kU/L	184/199	2.5 g egg protein 3 times/week from 4–6 months to 12 months vs. placebo until 12 months of age.	Positive egg s-IgE (≥0.35 kU/L). As secondary outcome: Prevalence of EA by OFC with raw egg powder at 1 year of age
STEP, Palmer 2017, Australia [53]	DBPCRCT	Egg (raw whole pasteurized)	High risk 4–6-month-old infants, maternal atopy history, no eczema, never taken egg	407/413	0.4 g egg protein/daily from 4–6 months to 10 months vs placebo; from 10 months, egg introduction liberalized in both groups.	Prevalence of EA by OFC with raw egg at 1 year of age
BEAT, Tan 2017, Australia [54]	DBPCRCT	Egg (raw whole pasteurized)	High risk 4-month-old infants, ≥ 1 relative (1st degree) with allergy	165/154	0.35 g egg protein/daily from 4 months to 8 months vs. placebo; from 8 months, egg introduction liberalized in both groups.	Positive SPT to egg white (≥3 mm). As secondary outcome: Prevalence of EA by OFC with lightly cooked egg at 1 year of age.
PETIT, Natsume 2017, Japan [55]	DBPCRCT	Egg (cooked lyophilized)	High risk 4–5 month-old infants with eczema, never taken egg	60/61	0.025 g egg protein/daily from 6 to 9 months, then 0.12 g from 9 to 12 months vs. placebo from 6 to 12 months.	Prevalence of EA by OFC with heated whole-egg powder at 1 year of age

BEAT, Beating Egg Allergy Trial; DBPC, Double-blind, placebo-controlled; EA, egg allergy; EAT, Enquiring About Tolerance; HEAP, Hen’s Egg Allergy Prevention; LEAP, Learning Early About Peanut Allergy; OFC, Oral Food Challenge; PEEAD, Preventing Peanut Allergy in Atopic Dermatitis; PETIT, Prevention of Egg Allergy with Tiny Amount Intake; RCT, Randomized Clinical Trial; STAR, Solid Timing for Allergy Research; STEP, Study Starting Time of Egg Protein; SPT, Skin prick test.

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
