# Peer review of "Prevention of Food Allergy: The Significance of Early Introduction"

_1010-660X, 2019, doi:10.3390/medicina55070323_

Round 1

Reviewer 1 Report

Comments:

The authors have written an excellent review on the latest RCT on timing of solid food introduction to infants as a means to prevent food allergy. The authors have presented an expert review which is easily understandable. I accept the review for publication with a minor revision.

Strengths of the review:

1.     The review summarizes the important recent advances made in food allergy prevention, compares the changes made to previous recommendations.

2.     The summary table of RCTs on Early Introduction of Allergenic Solids for Food Allergy Prevention is very informative and would be widely referenced.

Minor comments:

1.     The description of the development of tolerance vs allergy is a bit vague and I strongly suggest the authors to rephrase their views and make it clear especially the lines “The physiological development ….allergic responses.”

2.     In my opinion the description of Th2 response to food with IgE reaction as ‘antagonistic’ to ‘healthy’ IgG response is an oversimplified description, since presence of food specific IgE alone may not result in manifestation of food allergy.

3.     I suggest to the authors to include a short summary after each sub-section in the section 4 and provide your final comments if the early introduction of that particular food/form of the food is recommended. 

3.     In addition to the summarizing the results of the RCTs, I would encourage the authors to expand the discussion on the reasons why RCT on wheat or milk and other foods discussed in the review have not provided answers for/against early intervention to prevent food allergy.

Author Response

POINT-BY-POINT REPLY TO REVIEWERS

Article medicina-508727

We would like to thank you for your continued interest in our manuscript and for the opportunity to improve our manuscript. We have added all the information requested by the reviewer

REVIEWER 1 SPECIFIC COMMENTS

1. COMMENT: The description of the development of tolerance vs allergy is a bit vague and I strongly suggest the authors to rephrase their views and make it clear especially the lines “The physiological development ….allergic responses.”

RESPONSE: We agree. The pathogenesis of food allergy is a very complex topic, which goes beyond the purpose of this descriptive review. Considering the limited word count and the purpose of this review, we believe is it best to remove the paragraph from line 58 to 65, which would ultimately result a simplification of the topic and to refer the readers to more detailed reviews which focus on the advances and knowledge gaps in the pathophysiological mechanisms of food allergy. See lines 65-67

2.     In my opinion the description of Th2 response to food with IgE reaction as ‘antagonistic’ to ‘healthy’ IgG response is an oversimplified description, since presence of food specific IgE alone may not result in manifestation of food allergy.

RESPONSE: We agree. The pathogenesis of food allergy is a very complex topic, which goes beyond the purpose of this descriptive review. Considering the limited word count and the purpose of this review, we believe is it best to remove the paragraph from line 58 to 65, which would ultimately result a simplification of the topic and to refer the readers to more detailed reviews which focus on the advances and knowledge gaps in the pathophysiological mechanisms of food allergy. See lines 65-67

3.     I suggest to the authors to include a short summary after each sub-section in the section 4 and provide your final comments if the early introduction of that particular food/form of the food is recommended.

RESPONSE: We thank the Review for this comment. We included a key points right after the conclusion where we summarize the main findings of the 4 sections.

In regard to providing our final comments on the timing of introduction of specific foods to prevent food allergy, we have addressed this request in the Conclusion section, which has been modifies as follows (see Line 309-321): “although recent data suggest considering early introduction of allergenic solids as a potential strategy to tackle the rise in FA prevalence, convincing evidence for such practice is currently available only for peanut in high-risk infants between 4 and 11 months of life, but not for most other allergenic foods. At present, in line with current guidelines, we recommend a progressive introduction to solid foods, including all common allergenic solids, during the first year of life, according to the infant’s ability to chew, keep their head still and sit propped up, and familial or cultural habits, beginning at around 6 months but not before 4 months of age, possibly without discontinuing breastfeeding. For infants with severe AD and/or FA we recommend medical counselling before introducing common food allergens into the diet (i.e. peanut, egg, dairy and wheat and fiish), to exclude an IgE-sensitization to those foods, which would increase the risk of reaction upon ingestion, and discuss the best timing and modalities of such introductions in light of available evidence for FA prevention.”.

4.     In addition to the summarizing the results of the RCTs, I would encourage the authors to expand the discussion on the reasons why RCT on wheat or milk and other foods discussed in the review have not provided answers for/against early intervention to prevent food allergy.

RESPONSE: We thank the Review for this comment. We further discussed the reasons for lack of effective results for foods other than peanut and egg. See lines 246-262.

Please find attached the revised manuscript with tracked changes

Reviewer 2 Report

There have been quite a few recent studies on the same topic like: - https://aacijournal.biomedcentral.com/articles/10.1186/s13223-018-0286-1 - https://www.jacionline.org/article/S0091-6749(19)30001-6/pdf - https://link.springer.com/article/10.1007/s40521-014-0017-x How is this study different from these? What is the novelty aspect?

Author Response

We would like to thank you for your continued interest in our manuscript and for the opportunity to improve our manuscript. We have added all the information requested by the reviewer

REVIEWER 2 SPECIFIC COMMENTS

1. COMMENT: There have been quite a few recent studies on the same topic like. How is this study different from these? What is the novelty aspect?

·       Koplin J et al. Early Introduction of Foods for Food Allergy Prevention. Curr Treat Options Allergy;2014;1: 107. https://link.springer.com/article/10.1007/s40521-014-0017-x

·       Chan SE et al. Early introduction of foods to prevent food allergy. Allergy, Asthma & Clinical Immunology 2018;14(Suppl 2):57. https://aacijournal.biomedcentral.com/articles/10.1186/s13223-018-0286-1 

·       Bird JA, et al. Prevention of food allergy: Beyond peanut. J Allergy Clin Immunol 2019;143(2):545-547. https://www.jacionline.org/article/S0091-6749(19)30001-6/pdf 

RESPONSE: We due respect, this was an invited review on this specific topic. The reviewer is right we he states that there have been a few reviews on this topic, and they all try to summarize the most updated data. As the reviewer is fully aware this is an ongoing and active research topic, but this doesn’t mean that each review can quote new date, if new data have not been published yet. Regarding the specific 3 references quoted by the reviewer, we believe our study does provide new information, given that the :

- Koplin et al paper was published in 2014, long before the majority of the most recent randomized clinical trials on the topic would be published. Please check our Table 1, where 5 out 7 RCTs reported have been published after 2014;

- Chan SE et al. and Bird JA, et al, despite being very recent and updated commentary articles, they are mainly focused on peanut and egg, for which we currently have the most solid evidence. However, they do not give the same space to other allergenic foods as we tried to accomplish in our paper. Contrary to our paper, these two commentary articles do not provide neither a summary table of all RCTs on Early Introduction of Allergenic Solids for Food Allergy Prevention, nor an extensive discussion of how the changes to recommendations for solid food introduction in infants could impact breastfeeding practices. These two features can increase the interest of the readers and the readability of the paper. Finally, we included a key points box which can further contribute to the readability and citation of the paper.

Please find attached the revised manuscript with trached changes.

Round 2

Reviewer 2 Report

Satisfactory responses from the authors.